# Differential Involvement of Autophagy and Apoptosis in Response to Chemoendocrine and Endocrine Therapy in Breast Cancer: JBCRG-07TR

**DOI:** 10.3390/ijms20040984

**Published:** 2019-02-24

**Authors:** Takayuki Ueno, Norikazu Masuda, Shunji Kamigaki, Takashi Morimoto, Shigehira Saji, Shigeru Imoto, Hironobu Sasano, Masakazu Toi

**Affiliations:** 1Breast Surgical Oncology, Breast Oncology Center, Cancer Institute Hospital, Japanese Foundation for Cancer Research, Tokyo 135-8550, Japan; 2Department of Surgery, Breast Oncology, NHO Osaka National Hospital, Osaka 540-0006, Japan; nmasuda@alpha.ocn.ne.jp; 3Department of Breast Surgery Sakai Municipal Hospital, Osaka 593-8304, Japan; mitsu3880@yahoo.co.jp; 4Department of Breast Surgery Yao Municipal Hospital, Osaka 581-0069, Japan; mrmt@kcn.ne.jp; 5Department of Medical Oncology, Fukushima Medical University, Fukushima 960-1247, Japan; ss-saji@wa2.so-net.ne.jp; 6Department of Breast Surgery, Kyorin University Hospital, Tokyo 181-8611, Japan; imoto@ks.kyorin-u.ac.jp; 7Department of Pathology, Tohoku University School of Medicine, Sendai 103-0023, Japan; hsasano@patholo2.med.tohoku.ac.jp; 8Department of Surgery (Breast Surgery), Kyoto University, Kyoto 606-8507, Japan; toi@kuhp.kyoto-u.ac.jp

**Keywords:** beclin 1, LC3, TUNEL, M30, autophagy, apoptosis, metronomic, chemoendocrine therapy, endocrine therapy

## Abstract

Endocrine therapy is an essential component in the curative treatment of hormone receptor (HR)-positive breast cancer. To improve treatment efficacy, the addition of metronomic chemotherapy has been tested and shown to improve therapeutic effects. To better understand cellular reactions to metronomic chemoendocrine therapy, we studied autophagy-related markers, beclin 1 and LC3, and apoptosis-related markers, TUNEL and M30, in pre- and post-treatment cancer tissues from a multicenter neoadjuvant trial, JBCRG-07, in which oral cyclophosphamide plus letrozole were administered to postmenopausal patients with HR-positive breast cancer. Changes in the levels of markers were compared with those following neoadjuvant endocrine therapy according to clinical response. Apoptosis, in addition to autophagy-related markers, increased following metronomic chemoendocrine therapy and such increases were associated with clinical response. By contrast, following endocrine therapy, the levels of apoptosis-related markers did not increase regardless of clinical response, whereas the levels of autophagy-related markers increased. Furthermore, levels of the apoptosis-related marker, M30, decreased in responders of endocrine therapy, suggesting that the induction of apoptosis by metronomic chemoendocrine therapy was involved in the improved clinical outcome compared with endocrine therapy. In conclusion, metronomic chemoendocrine therapy induced a different cellular reaction from that of endocrine therapy, including the induction of apoptosis, which is likely to contribute to improved efficacy compared with endocrine therapy alone.

## 1. Introduction

Endocrine therapy is one of the essential components of curative treatment for hormone receptor (HR)-positive early stage breast cancer. Even following a five-year course of endocrine treatment, certain patients experience cancer recurrence that continues steadily [1]. In order to improve the treatment efficacy and prognosis of patients, a number of strategies have been proposed, including the extension of treatment duration, the addition of molecular target therapy such as an mTOR inhibitor and a cyclin-dependent kinase (CDK) 4/6 inhibitor, and the addition of metronomic chemotherapy [2,3,4,5].

Metronomic chemotherapy is the repeated administration of antineoplastic drugs at low doses frequently to avoid toxic side effects [6,7]. Metronomic chemotherapy has been suggested to induce anti-cancer effects via multiple mechanisms, including anti-angiogenesis, anti-tumor immune response, and direct anti-cancer action [8]. Low-dose oral cyclophosphamide is one of the most commonly administered metronomic agents either alone or together with other agents such as capecitabine and methotrexate [9,10,11,12]. Because metronomic chemotherapy induces anti-cancer effects with minimal side effects, it is a good candidate for combined use with endocrine therapy.

We have previously shown that the addition of metronomic chemotherapy, an oral cyclophosphamide, to endocrine therapy increases breast-conservation rate with minimal toxicity [13]. In addition, we reported that a response-guided approach, such as the addition of metronomic cyclophosphamide in patients not responding to an 8–12-week course of endocrine therapy, resulted in a tumor response comparable to that in responders who were treated with endocrine therapy alone [14]. Therefore, the addition of metronomic chemotherapy to endocrine therapy is a promising strategy for improving patient outcomes without additional major toxicity.

In general, anticancer therapy induces either cytocidal or cytostatic effects on cancer cells. Endocrine therapy has been shown to induce cytostatic effects [15]. However, as endocrine therapy reduces cancer volume in almost 70% of cases of HR-positive early breast cancer, it also induces cytocidal effects, although the mode of cell death is not clear [16]. We and others have shown that endocrine therapy does not induce apoptosis but that it does induce autophagy in HR-positive breast cancer [16,17]. Therefore, autophagy is likely to be involved in the reduction of tumor volume by endocrine therapy.

In order to improve treatment efficacy and to overcome treatment resistance, it is critical to understand how cancer cells react to anticancer treatments. As the mode of cell death involved in the tumor response to metronomic chemotherapy with endocrine therapy (chemoendocrine therapy) remains to be elucidated, we studied autophagy-related markers, beclin 1 and LC3, and apoptosis-related markers, TUNEL and M30, in archived breast cancer tissue specimens from a prospective multicenter neoadjuvant chemoendocrine trial, JBCRG-07, in which an oral cyclophosphamide was administered together with an aromatase inhibitor, letrozole, in postmenopausal patients with HR-positive breast cancer. We compared the clinical response with changes in the levels of the markers and found that apoptosis and autophagy were involved in the clinical response to metronomic chemoendocrine therapy while autophagy, but not apoptosis, was involved in the response to endocrine therapy.

## 2. Results

### 2.1. Induction of Autophagy and Apoptosis by Metronomic Chemoendocrine Therapy

Tissue samples were collected from 38 of the 41 patients in the JBCRG-07 study. The baseline characteristics of the 38 patients are shown in Table 1.

Autophagy was examined by evaluating levels of beclin 1 and LC3 and apoptosis was examined by evaluating levels of TUNEL and M30 in tissues from the neoadjuvant letrozole and low-dose cyclophosphamide study (JBCRG-07). Representative images for each marker are shown in Figure 1A.

Both of the autophagy-related markers increased following metronomic chemoendocrine therapy (Figure 1B, *p* = 0.0010 and 0.0030 for beclin 1 and LC3, respectively). Similarly, the two apoptosis-related markers increased following treatment (Figure 1B, *p* = 0.0053 and 0.0006 for TUNEL and M30, respectively).

### 2.2. Association of Autophagy- and Apoptosis-Related Markers with Clinical Response to Metronomic Chemoendocrine Therapy

The association between changes in the levels of each marker and the clinical response to metronomic chemoendocrine therapy was examined. The baseline characteristics according to clinical response are shown in Appendix A. The autophagy-related markers beclin 1 and LC3 increased significantly in the responders (Figure 2A, *p* = 0.012 and 0.043, respectively) but not in the non-responders. Similarly, the apoptosis-related marker M30 increased significantly in the responders (Figure 2B, *p* = 0.0059) but not in the non-responders. TUNEL showed a trend for increase in the responders (Figure 2B, *p* = 0.060).

### 2.3. Association between the Levels of Autophagy- and Apoptosis-Related Markers and the Clinical Response to Endocrine Therapy

In our previous study, we showed that endocrine therapy induces autophagy but not apoptosis [17]. In the present study, we investigated whether autophagy or apoptosis were associated with the clinical response to endocrine therapy using samples from the multicenter neoadjuvant exemestane trial (JFMC34-0601). The baseline characteristics of patients in JFMC34-0601 according to clinical response are shown in Appendix A. The levels of autophagy-related markers, beclin 1 and LC3, were increased significantly in patients who showed a clinical response to endocrine therapy (Figure 3A, *p* = 0.022 and 0.020, respectively). LC3 also increased in the non-responders (*p* = 0.016). The apoptosis-related markers, TUNEL and M30, did not increase in either the responders or the non-responders (Figure 3B), although M30 decreased in the responders (Figure 3B, *p* = 0.014).

### 2.4. Association of Autophagy- and Apoptosis-Related Markers with Patients’ Survival

The association between the pre-treatment levels of each marker and the patients’ survival was examined in JBCRG07 study. Disease-free survival (DFS) showed a trend for association with either of apoptosis-related markers with no statistical significance (*p* = 0.09 for both TUNEL and M30) while no association was observed between DFS and autophagy-related markers (Figure 4a). Overall survival (OS) was significantly associated with the apoptosis-related marker M30 (*p* = 0.045) but not with the other markers (Figure 4b). Patients with the positive expression of pre-treatment M30 showed a significantly poorer OS than those with the negative expression.

## 3. Discussion

In this study, we demonstrated that metronomic chemoendocrine therapy increased the expression of autophagy-related markers, beclin 1 and LC3, and the expression of apoptosis-related markers, TUNEL and M30, in HR-positive breast cancer tissues. In addition, such increases were more evident in the responders than in the non-responders. This was different from the cellular response to endocrine therapy as endocrine therapy did not increase the expression of the apoptosis-related markers and decreased the expression of M30 in the responders. Therefore, the induction of apoptosis by metronomic chemoendocrine therapy explains, at least in part, the improved efficacy of chemoendocrine therapy compared with endocrine therapy alone [13,14]. In this regard, a response-guided approach in which metronomic chemotherapy is added to the treatment of non-responders to short-term exposure of endocrine therapy is a reasonable option to improve treatment efficacy [14].

Autophagy is a mechanism of cell death and our results that both autophagy-related markers increased following chemoendocrine and endocrine therapy, particularly in the responders, suggest that autophagy is one of the mechanisms that reduce tumor volume in HR-positive breast cancer. However, some reports show contrasting results that the inhibition of autophagy sensitizes cancer cells to aromatase inhibitors [18,19]. Therefore, autophagy may be involved in both endocrine response and endocrine resistance. It is of clinical importance to investigate whether autophagy in the remaining cancer cells following treatment results in cancer cell death or survival. This can be determined by examining autophagy in tissues from the middle of the treatment and comparing these with the clinical response following treatment; this should be performed in a future study.

Clarifying the mode of cell death by a certain treatment will be useful in understanding resistance mechanisms. In the present study, apoptosis-related markers, TUNEL and M30, did not increase, whereas M30 decreased in responders to exemestane, suggesting that resistance to apoptosis is not directly associated with resistance to exemestane. The reason why apoptosis decreased following endocrine therapy is not clear, however; two neoadjuvant endocrine studies showed consistent results [16,17]. One possible explanation is that a high apoptotic index has been shown to correlate with a high proliferation index in breast cancer tissues [20,21,22] and so the cytostatic effect of endocrine therapy leads to reduced cancer proliferation, resulting in a lower apoptotic index in cancer cells.

In this study, we used beclin 1 and LC3 for autophagy-related markers and TUNEL and M30 for apoptosis-related markers. Beclin 1 is a coiled-coil myosin-like BCL2-interacting protein that is involved in initiation and nucleation of the phagophore, whereas LC3 is a ubiquitin-like protein that is involved in elongation and closure of the autophagosome [23]. Thus, we considered beclin 1 and LC3 suitable for autophagy detection because of involvement in different autophagic phases and different turnovers. There are other markers for autophagy, including p62, which is involved in autophagy-dependent elimination of different cargos including ubiquitinated protein aggregates and should be considered for future studies [24]. TUNEL stands for terminal deoxynucleotidyl transferase (TdT) dUTP Nick-End Labeling and detects apoptotic cells with extensive DNA degradation during the late stage of apoptosis [25], while M30 detects fragments of cytokeratin-18 cleaved by caspases during apoptosis and is shown to correlate with the expression of activated caspase-3 [26,27,28]. M30 is suitable for apoptosis detection in cancer cells because cytokeratin-18 is expressed in epithelial cells and apoptosis in non-epithelial cells including stromal cells can be excluded by M30. Additional autophagy- and apoptosis-related markers will be useful in future studies to elucidate more detailed mechanisms underlying drug-induced cell death by different anti-cancer agents.

We showed that pre-treatments levels of apoptosis-related markers showed a trend for association with DFS and that M30 was significantly associated with overall survival (OS), which is in concordance with the previous reports showing that apoptotic index was associated with aggressive phenotypes of breast cancer and unfavorable prognosis [29]. On the other hand, autophagy-related markers did not show any association with survival in the present study, indicating that autophagy-related markers do not have as clear an impact on survival as apoptosis-related markers. In fact, the prognostic impact of autophagy-related markers is controversial [30] and may differ for different biological subtypes of breast cancer. Further larger studies are necessary to clarify the prognostic value of autophagy-related markers.

There were a number of limitations in the present study. First, the sample sizes of both the chemoendocrine study (JBCRG-07) and endocrine study (JFMC34) were small. Therefore, the lack of significant changes in the levels of autophagy- or apoptosis-related markers in the non-responders may have been due to a small sample size and, thus, requires cautious interpretation. Secondly, a limited number of markers related to autophagy and apoptosis were examined. It is necessary to include other markers, for example p63, to determine more precisely the role of autophagy and apoptosis in response to anticancer therapy in a future study. Thirdly, there are additional modes of cell death, other than autophagy and apoptosis. It is clinically important to investigate other cell death modes, including senescence and mitotic catastrophe, using tissues prior to and following different types of anticancer treatment. Another limitation is that only oral cyclophosphamide was tested as a metronomic therapy. Therefore, it is not clear how other metronomic drugs, including oral 5-fluorouracil, cause anticancer effects in cancer cells. It will be useful to elucidate the cellular responses to different metronomic drugs in order to identify a suitable combination for use with endocrine treatment.

In conclusion, we demonstrated that metronomic chemoendocrine therapy with oral cyclophosphamide induced apoptosis and autophagy in cancer tissues, whereas endocrine therapy did not induce apoptosis. The difference in the cellular response to different therapies helps to explain the difference in clinical outcomes and supports improved efficacy by the response-guided addition of metronomic therapy to endocrine therapy [14]. Further studies are warranted to investigate which mode of cell death is involved in the anticancer effects of different metronomic drugs to consider an optimal treatment strategy for endocrine therapy.

## 4. Materials and Methods

### 4.1. Clinical Trials

The design of the clinical trial, JBCRG-07, is described elsewhere (Registration number: UMIN 000001331) [13]. Briefly, patients with T2-4 N0-1 and estrogen receptor (ER)-positive breast cancer were enrolled between October 2007 and March 2010. ER-positivity was defined by ≥10% nuclear staining. Patients received 2.5 mg/day letrozole and 50 mg/day oral cyclophosphamide daily for 24 weeks, and surgery was performed 1–4 weeks following the final administration. The multicenter prospective neoadjuvant exemestane study, JFMC34-0601, was a single-arm phase II study of neoadjuvant endocrine therapy (Registration number: UMIN C000000345). The design of the trial is described elsewhere [31,32,33,34]. Briefly, the eligibility criteria included postmenopausal women aged 55–75 years with stage II or IIIa invasive breast cancer with positive ER status (≥10% nuclear staining). Patients received 25 mg/day exemestane for 16 weeks followed by an 8-week extension depending on the treatment response. Surgery was performed at 24 weeks. Patients with progressive disease were withdrawn from the study and offered appropriate treatment, including surgery and other anticancer drugs. Clinical response was assessed by investigators by combining caliper measurements and images obtained via ultrasound, computed tomography, and Magnetic Resonance Imaging according to the Response Evaluation Criteria in Solid Tumors version 1.0 [32]. Written informed consent was obtained from all patients who participated in the studies. The studies conform to the provisions of the Declaration of Helsinki.

### 4.2. Immunohistochemistry (IHC)

The IHC staining was performed using a Histofine Kit (Nichirei, Tokyo). The positivity of ER and progesterone receptor (PgR) was defined by ≥10% nuclear staining. The expression of human epidermal growth factor receptor 2 (HER2) was examined using the HercepTest (Dako, Tokyo). HER2-positivity was defined as either 3+ or 2+ with *HER2* gene amplification by fluorescence in situ hybridization. The autophagy- and apoptosis-related markers were stained using the anti-beclin 1 antibody (1:250; NB500-249; Novus Biologicals, CO, USA), anti-LC3 antibody (1:200; PM036; MBL, Nagoya, Japan), TUNEL (In Situ Cell Death Detection Kit; Roche Diagnostics, Mannheim, Germany) and M30 CytoDEATH (1:100; Roche Diagnostics). The cytoplasmic staining of beclin 1, LC3, and M30 and nuclear staining of TUNEL were assessed with pre- and post-treatment tissues. The expression rate of each marker was assessed as positive cancer cells per total cancer cells.

### 4.3. Statistical Analysis

Statistical analyses were performed using the Wilcoxon’s paired test for comparisons between pre- and post-treatment levels of autophagy- and apoptosis-related markers. Overall survival (OS) and disease-free survival (DFS) were estimated and compared using the Kaplan–Meier method and log-rank test between groups. All analyses were performed using the JMP Ver.13.2.1 (SAS Institute, Inc., Cary, NC, USA). All P values were two-sided, and *p* < 0.05 was considered statistically significant. All graphs were produced using the GraphPad Prism ver. 7 (GraphPad Software, San Diego, CA, USA).

## Figures and Tables

**Figure 1 ijms-20-00984-f001:**
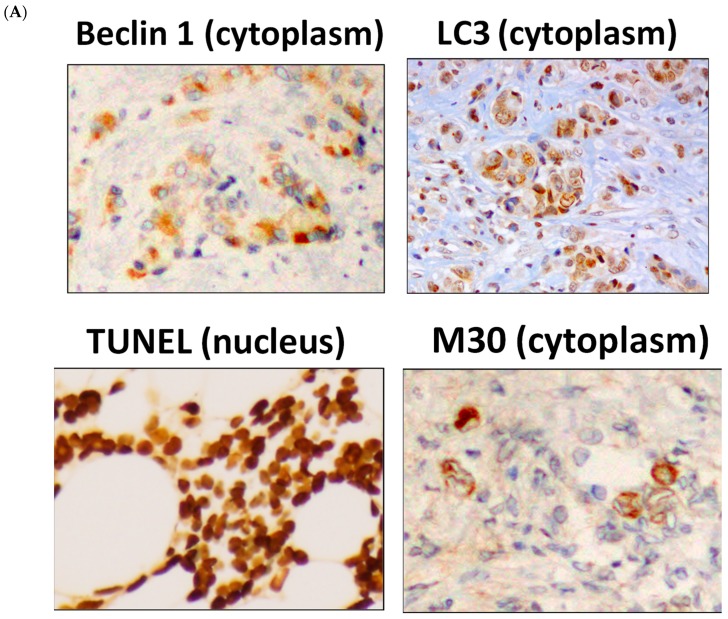
Increases in autophagy- and apoptosis-related markers following metronomic chemoendocrine therapy. (**A**) Images for autophagy-related markers, beclin 1 and LC3, and apoptosis-related markers, TUNEL and M30, in breast cancer tissues (Scale bar = 50 µm). (**B**) Both autophagy-related markers and both apoptosis-related markers increased significantly following metronomic chemoendocrine treatment. A solid upward arrow indicates a significant increase following treatment.

**Figure 2 ijms-20-00984-f002:**
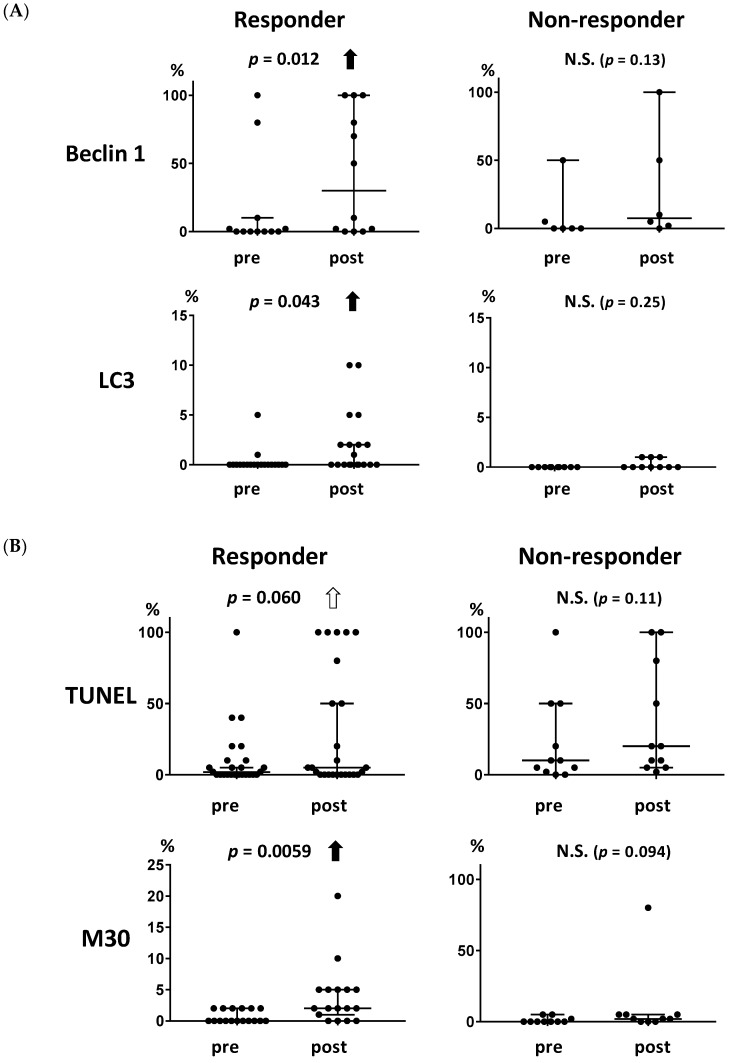
Association between the levels of autophagy- and apoptosis-related markers and the clinical response to metronomic chemoendocrine therapy. (**A**) Both autophagy-related markers, beclin 1 and LC3, increased significantly in the responders (*p* = 0.012 and 0.043, respectively) but not in the non-responders. (**B**) Apoptosis-related marker M30 increased in the responders (*p* = 0.0059) but not in the non-responders. Similarly, TUNEL showed a trend for increase in the responders (*p* = 0.060). A solid upward arrow indicates a significant increase following treatment. An open upward arrow indicates a trend for increase following treatment. N.S. not significant.

**Figure 3 ijms-20-00984-f003:**
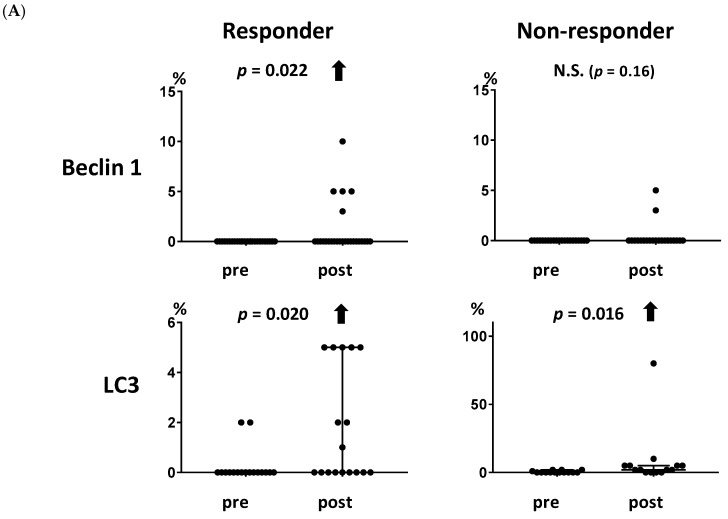
Association between the levels of autophagy- and apoptosis-related markers and the clinical response to endocrine therapy. (**A**) Both autophagy-related markers, beclin 1 and LC3, increased significantly in the responders (*p* = 0.022 and 0.020, respectively). LC3 also increased in the non-responders (*p* = 0.016). (**B**) Neither of the apoptosis-related markers, TUNEL and M30, increased in the responders or non-responders. M30 decreased significantly in the responders (*p* = 0.014). A solid upward arrow indicates a significant increase following treatment. A solid downward arrow indicates a significant decrease following treatment. N.S. not significant.

**Figure 4 ijms-20-00984-f004:**
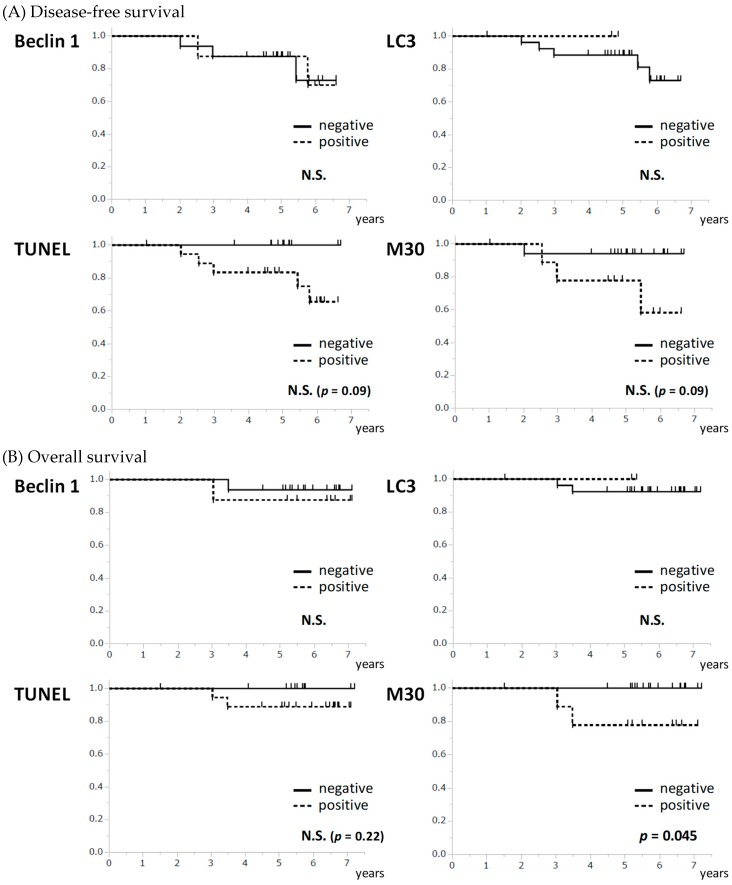
Association between the pre-treatment levels of autophagy- and apoptosis-related markers and survival. (**A**) None of the apoptosis-related and autophagy-related markers showed an association with disease-free survival. (**B**) M30 was significantly associated with overall survival (*p* = 0.045) although none of the other markers showed an association. N.S. not significant.

**Table 1 ijms-20-00984-t001:** Baseline characteristics of patients.

Characteristics	Subgroup	Number
Number of patients		38
Average age (range)		69.7 (57–82)
T	T1	1
	T2	33
	T3	4
N	N0	33
	N1	5
ER	+	38
	−	0
PgR	+	24
	−	14
HER2	+	8
	−	30
Histological grade	1	13
	2	25
	3	0

ER, estrogen receptor; PgR, progesterone receptor; HER2, human epidermal growth factor receptor 2. T, tumor, N, nodal status.

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
