# Peer review of "Differential Involvement of Autophagy and Apoptosis in Response to Chemoendocrine and Endocrine Therapy in Breast Cancer: JBCRG-07TR"

_ijms, 2019, doi:10.3390/ijms20040984_

Round 1

Reviewer 1 Report

The authors addressed my questions, I don’t have additional comments.

Reviewer 2 Report

The authors have made the changes as requested. 

This manuscript is a resubmission of an earlier submission. The following is a list of the peer review reports and author responses from that submission.

Round 1

Reviewer 1 Report

This is a great work related to the markers of auto phage and apoptosis in response of two types of therapy in breast cancer. 

I only have a few questions for the authors:

I understand why you chose these specific autophagy markers, but why did you choose M30 and Tunel as apoptosis markers? Why not Effector caspases, like caspase 3 or 7? 

Related to the lack of apoptosis markers on patients treated with endocrine treatment, I agree that maybe it will be due to the cells are in an autophagy stage, reducing the tumor volume. As a reminder, autophagy is induced in the cell because of a stress, if the cell cannot handle this stress, the signaling changes to apoptosis. Will it be possible for you with your samples to determine if the cells were actually in an autophagy stage? Like using SEM? 

Please consider as an asset for your work test additional apoptosis markers. 

Thank you very much. 

Author Response

I only have a few questions for the authors: I understand why you chose these specific autophagy markers, but why did you choose M30 and Tunel as apoptosis markers? Why not Effector caspases, like caspase 3 or 7? 

Response: Thank you very much for this important comment. We chose TUNEL and M30 because these markers detect apoptosis during different stages of apoptosis and, in particular, M30 detects fragments of cytokeratin-18 which is cleaved by caspases and expressed only in epithelia. These markers have been shown to correlate with activated caspase-3 expression (Duan et al. 2003). We have added discussion on the choice of the markers in Discussion.

“In this study, we used beclin 1 and LC3 for autophagy-related markers and TUNEL and M30 for apoptosis-related markers. Beclin 1 is a coiled-coil myosin-like BCL2-interacting protein that is involved in initiation and nucleation of the phagophore, whereas LC3 is a ubiquitin-like protein that is involved in elongation and closure of the autophagosome [23]. Thus, we considered beclin 1 and LC3 suitable for autophagy detection because of involvement in different autophagic phases and different turnovers. There are other markers for autophagy, including p62, which is involved in autophagy-dependent elimination of different cargos including ubiquitinated protein aggregates and should be considered for future studies [24]. TUNEL stands for terminal deoxynucleotidyl transferase (TdT) dUTP Nick-End Labeling and detects apoptotic cells with extensive DNA degradation during the late stage of apoptosis[25] while M30 detects fragments of cytokeratin-18 cleaved by caspases during apoptosis and is shown to correlate with the expression of activated caspase-3 [26-28]. M30 is suitable for apoptosis detection in cancer cells because cytokeratin-18 is expressed in epithelial cells and apoptosis in non-epithelial cells including stromal cells can be excluded by M30.”

Related to the lack of apoptosis markers on patients treated with endocrine treatment, I agree that maybe it will be due to the cells are in an autophagy stage, reducing the tumor volume. As a reminder, autophagy is induced in the cell because of a stress, if the cell cannot handle this stress, the signaling changes to apoptosis. Will it be possible for you with your samples to determine if the cells were actually in an autophagy stage? Like using SEM? 

Response: We agree that SEM is very useful and reliable for detection of autophagy. However, SEM is not available in our laboratory and we will consider future studies using SEM by collaborating with another laboratory.

Please consider as an asset for your work test additional apoptosis markers. 

Response: We agree that additional markers will help to elucidate more detailed mechanisms of drug-induced cell death. We have discussed this point in Discussion.

“Additional autophagy- and apoptosis-related markers will be useful in future studies to elucidate more detailed mechanisms underlying drug-induced cell death by different anti-cancer agents.”

Reviewer 2 Report

The goal of this manuscript is to demonstrate different responses in terms of apoptosis and autophagy following metronomic chemotherapy compared to those following neoadjuvant endocrine therapy. The authors show by IHC that markers for apoptosis or autophagy are differentially increased between responders and non-responders in patients enrolled in the clinical trial JBCRG-07TR.

The results here presented build up in a recent paper of the research group regarding the clinical trial JBCRG-07TR in which neoadjuvant letrozole plus cyclophosphamide showed a good clinical response for postmenopausal patients with estrogen receptorpositive breast cancer. The novelty presented is in terms of the molecular effects induced in breast cancer cells of patients undergoing metronomic chemoendocrine therapy and the relation the clinical response.

The paper is technically sound and the work appears to be carefully carried out and conclusions drawn of interest. The paper evidences clearly different responses of the selected markers between both studies; however, with exception of M30, none of them was associated with DFS or OS with statistical significance. I recommend that the paper is published  after the following questions are addressed. 

       The concept of metronomic chemotherapy is becoming increasingly accepted, having demonstrated over the years in several clinical trials beneficial effects in particular for the most aggressive/metastatic subtypes of breast cancer, particularly in the lower cytotoxicity induced. The authors should include some references in their Introduction section for other studies or recent reviews demonstrating the impact of metronomic chemotherapy in clinical responses.  

The authors should clarifiy the choice of the markers, in particular because it is declared in the Discussion section that other markers could have been used for autophagy. 

In Figure 4, panel A the P-values for TUNEL and M30 should be stated as N.S. and the corresponding value should be presented parentheses, as in the remaining figures. In the Figure legend the associations should also be declared as N.S.  

Author Response

The paper is technically sound and the work appears to be carefully carried out and conclusions drawn of interest. The paper evidences clearly different responses of the selected markers between both studies; however, with exception of M30, none of them was associated with DFS or OS with statistical significance. I recommend that the paper is published after the following questions are addressed. 

Response: Thank you very much for your careful review and important comments.

       The concept of metronomic chemotherapy is becoming increasingly accepted, having demonstrated over the years in several clinical trials beneficial effects in particular for the most aggressive/metastatic subtypes of breast cancer, particularly in the lower cytotoxicity induced. The authors should include some references in their Introduction section for other studies or recent reviews demonstrating the impact of metronomic chemotherapy in clinical responses.  

Response: Thank you for the constructive comment. We added description on metronomic chemotherapy with some references to explain the clinical relevance of metronomic chemotherapy in the Introduction.

 “Metronomic chemotherapy is the repeated administration of antineoplastic drugs at low doses frequently to avoid toxic side effects [6,7]. Metronomic chemotherapy has been suggested to induce anti-cancer effects via multiple mechanisms including anti-angiogenesis, anti-tumor immune response, and direct anti-cancer action [8]. Low-dose oral cyclophosphamide is one of the most commonly administered metronomic agents either alone or together with other agents such as capecitabine and methotrexate [9-12]. Because metronomic chemotherapy induces anti-cancer effects with minimal side effects, it is a good candidate for combined use with endocrine therapy.

The authors should clarify the choice of the markers, in particular because it is declared in the Discussion section that other markers could have been used for autophagy. 

Response: We have addressed this issue in Discussion to clarify the choice of the markers.

“In this study, we used beclin 1 and LC3 for autophagy-related markers and TUNEL and M30 for apoptosis-related markers. Beclin 1 is a coiled-coil myosin-like BCL2-interacting protein that is involved in initiation and nucleation of the phagophore, whereas LC3 is a ubiquitin-like protein that is involved in elongation and closure of the autophagosome [23]. Thus, we considered beclin 1 and LC3 suitable for autophagy detection because of involvement in different autophagic phases and different turnovers. There are other markers for autophagy, including p62, which is involved in autophagy-dependent elimination of different cargos including ubiquitinated protein aggregates and should be considered for future studies [24]. TUNEL stands for terminal deoxynucleotidyl transferase (TdT) dUTP Nick-End Labeling and detects apoptotic cells with extensive DNA degradation during the late stage of apoptosis[25] while M30 detects fragments of cytokeratin-18 cleaved by caspases during apoptosis and is shown to correlate with the expression of activated caspase-3 [26-28]. M30 is suitable for apoptosis detection in cancer cells because cytokeratin-18 is expressed in epithelial cells and apoptosis in non-epithelial cells including stromal cells can be excluded by M30.”

In Figure 4, panel A the P-values for TUNEL and M30 should be stated as N.S. and the corresponding value should be presented parentheses, as in the remaining figures. In the Figure legend the associations should also be declared as N.S.  

Response: The P-values have been stated as N.S. with the corresponding value in parentheses. The figure legends have also been changed in accordance with the figure.